# Editing the *RR-TZF* Gene Subfamily in Rice Uncovers Potential Risks of CRISPR/Cas9 for Targeted Genetic Modification

**DOI:** 10.3390/ijms26031354

**Published:** 2025-02-05

**Authors:** Shufen Zhou, Dagang Tian, Huaqing Liu, Xiaozhuan Lu, Di Zhang, Rui Chen, Shaohua Yang, Weiren Wu, Feng Wang

**Affiliations:** 1Key Laboratory of Genetics, Breeding and Multiple Utilization of Crops, Ministry of Education, Fujian Provincial Key Laboratory of Crop Breeding by Design, Fujian Agriculture and Forestry University, Fuzhou 350002, China; zsf@fjage.org (S.Z.); lxz2025@163.com (X.L.); 2Fujian Key Laboratory of Genetic Engineering for Agriculture, Biotechnology Research Institute, Fujian Academy of Agricultural Sciences, Fuzhou 350003, China; tdg@fjage.org (D.T.); lhq@fjage.org (H.L.); zhangd202501@163.com (D.Z.); cr@fjage.org (R.C.); ysh@fjage.org (S.Y.)

**Keywords:** CRISPR/Cas9, *RR-TZF* gene subfamily, rice (*Oryza sativa* L.), off-target effects, inheritable mutagenicity

## Abstract

The CRISPR/Cas9 system offers a powerful tool for gene editing to enhance rice productivity. In this study, we successfully edited eight *RR-TZF* genes in *japonica* rice Nipponbare using CRISPR-Cas9 technology, achieving a high editing efficiency of 73.8%. Sequencing revealed predominantly short insertions or deletions near the PAM sequence, along with multi-base deletions often flanked by identical bases. Off-target analysis identified 5 out of 31 predicted sites, suggesting the potential for off-target effects, which can be mitigated by designing gRNAs with more than three base mismatches. Notably, new mutations emerged in the progeny of several gene-edited mutants, indicating inheritable genetic mutagenicity. Phenotypic analysis of homozygous mutants revealed varied agronomic traits, even within the same gene, highlighting the complexity of gene-editing outcomes. These findings underscore the importance of backcrossing to minimize off-target and inheritable mutagenicity effects, ensuring more accurate trait evaluation. This study offers insights into CRISPR/Cas9 mechanisms and uncertain factors and may inform future strategies for rice improvement, prompting further research into CRISPR/Cas9’s precision and long-term impacts.

## 1. Introduction

Rice (*Oryza sativa* L.), a staple food for more than half of the world’s population, is crucial for global food security. However, the increasing population and deteriorating environmental conditions pose challenges to improving rice yield [1,2]. Traditional methods of rice breeding have reached a plateau [3,4], underscoring the pressing need for more advanced technological interventions. The advent of the clustered regularly interspaced short palindromic repeat/CRISPR-associated protein 9 (CRISPR/Cas9) system has revolutionized the field of genetic engineering, offering a powerful tool for targeted gene modification and crop enhancement [5,6]. This technology holds the promise of enhancing rice productivity by precisely manipulating genes involved in growth, development, and stress responses [7,8].

The CRISPR/Cas9 system uses small guided RNA molecules to direct the Cas9 nuclease to recognize and cut target DNA sequences, where it induces double-strand breaks (DSBs) [9]. The DSBs can be mended by the cell’s repair machinery including the error-prone nonhomologous end joining (NHEJ) pathway and the more accurate homology-directed repair (HDR) pathway [10]. During cleavage site repairing, the NHEJ pathway often introduces small InDels (insertions and deletions) or substitutions and leads to functional changes in the target gene, which can be utilized to elucidate gene functions and to develop novel germplasm resources [11]. Currently, CRISPR/Cas9 technology has been widely applied across various crops to improve traits [12,13]. In rice, it has been successfully utilized to improve resistance to biotic and abiotic stresses, as well as develop new germplasms with improved nutritional content and agronomic performance [14,15,16,17,18]. These advancements have the potential to address critical agricultural challenges, such as food security and crop resilience in the face of environmental stress. However, the quest for editing accuracy and the generation of complex mutations remain significant considerations in the practical application of this technology [19,20].

The *Arginine-rich tandem CCCH zinc finger* (*RR-TZF*) gene subfamily, unique to plants, is characterized by the presence of an arginine-rich (RR) region and a tandem CCCH zinc finger (TZF) motif. As plant genome sequencing projects continue to escalate, the *RR-TZF* gene subfamily has been systematically identified across various plant species, such as 11 members in *Arabidopsis thaliana*, 9 in rice (*Oryza sativa*), and 12 in maize (*Zea mays*) [21]. These genes play crucial roles in development and stress responses, including but not limited to seed germination, flowering time, and responses to biotic and abiotic stresses [22]. In rice, *OsTZF1* is linked to seed germination, seedling growth, leaf senescence, and oxidative stress tolerance [23,24]. *OsTZF2* (*OsDOS*) delays MeJA-induced leaf senescence [25]. *OsTZF5* and *OsTZF7* enhance drought response [26,27], whereas *OsTZF8* (*OsC3H10*) is highly expressed in seeds, significantly improving drought tolerance [28]. Lastly, *OsGZF1* (*OsTZF9*) regulates the *GluB-1* promoter, influencing glutelin protein accumulation during grain development [29]. This highlights the *RR-TZF* gene subfamily’s crucial role in rice growth, stress resistance, and grain quality.

Given their significance in fundamental plant functions, the *RR-TZF* gene subfamily emerges as a crucial target for CRISPR/Cas9-mediated crop improvement, offering promising avenues for agricultural innovation and food security. In this study, we employed the CRISPR/Cas9 gene-editing system to target nine *RR-TZF* genes in rice. Our objectives included characterizing the induced mutations, assessing the potential for off-target effects, evaluating the transmission of these mutations across generations, and examining the phenotypic manifestations of these mutations under normal growth conditions, with a particular focus on the implications of genetic variability induced by CRISPR/Cas9 for functional genomics and future rice improvement strategies.

## 2. Results

### 2.1. CRISPR/Cas9-Mediated Mutagenesis in Rice RR-TZF Genes

To enhance effective editing, we selected one to three target sites within the coding regions of the nine rice *RR-TZF* genes for CRISPR/Cas9 editing (Appendix A). The sgRNAs were synthesized and integrated into the CRISPR/Cas9 binary vector, yielding 12 recombinant vectors for *Agrobacterium*-mediated transformation of the calli of Nipponbare. Notably, certain samples were subjected to dual-vector co-infection to obtain double-mutant combinations (Table 1).

Eleven of the twelve vectors successfully yielded regenerated plants, with the exception being the vector targeting KTZF9a. PCR assays and Sanger sequencing confirmed the presence of Cas9/gRNA transgenes in 328 lines, of which 211 were derived from co-infection experiments (Table 1). Among these, 16 plants from six co-infection groups were identified as co-transformed, with a co-transformation rate varying from 3.9% to 13.8% (Table 1). This suggests that co-infection can result in a modest proportion of double transformations.

Direct sequencing of PCR products spanning the predicted target sites in all T_0_ transgenic plants revealed 219 mutated lines, including 9 double mutants. Interestingly, none of the 35 transgenic lines targeted for KTZF3 showed mutations, possibly influenced by sequence specificity or chromosomal positioning. Mutation rates across the remaining 10 target sites varied from 43.8% to 90.9%, with an average of 73.8% (Table 2). The nine double mutants exhibited various mutation combinations, with 56.3% being a result of double mutagenesis (Table 1). These results indicate high efficiency in inducing targeted mutagenesis for all constructs except that for targeting KTZF3.

Sequencing data allowed us to genotype all mutants, categorizing mutations into homozygous, heterozygous, bi-allelic, and chimeric types (Table 2). Homozygous mutations were identified across T_0_ plants for all ten vectors, with detection rates ranging from 8.6% to 76.5%. Bi-allelic mutations were observed in T_0_ plants for nine vectors, with the highest rate of up to 68.2%. Heterozygous and chimeric mutations were also identified in T_0_ plants from seven vectors, with a detection rate of 10.7% and 4.9%, respectively. In total, the CRISPR/Cas9-mediated editing successfully established a library of 219 *RR-TZF* mutants, providing a rich resource for genomic modifications.

### 2.2. Diversity of Targeted Mutations Induced by CRISPR/Cas9 Editing

Our comprehensive sequence analysis of 416 CRISPR/Cas9-edited target sites within the rice genome revealed a predominance of insertions (43.8%) and deletions (48.6%), with a minor occurrence of substitutions (0.2%) and complex mutations (7.4%) that included at least two types of insertions, deletions, or substitutions (Figure 1A and Appendix A). Notably, both insertions and deletions were observed across all effective target sites, with KTZF 5 and KTZF 9c exhibiting the highest mutation rates for insertions and deletions, respectively, at 86.4% and 86.2% (Figure 1B and Appendix A). The insertions identified were exclusively single-base, while deletions ranged from one-base to multi-base, demonstrating the versatility of the CRISPR/Cas9 system in inducing different types of mutations (Figure 1A). Strikingly, small-fragment deletions of less than 10 base pairs were observed more frequently than larger-fragment deletions (Figure 1A). The findings indicated a propensity for the generation of short indels at the targeted sites.

The spatial distribution of mutations relative to the PAM site was also scrutinized (Figure 2). The majority of single-base-pair (1 bp) insertions (180 out of 182) occurred at the fourth base from the PAM site, with a preponderance of adenine (A) or thymine (T) (Figure 2A). Conversely, single-base-pair deletions were chiefly located at the fourth or fifth base from the PAM site, with guanine (G) or cytosine (C) more frequently deleted compared to A or T (Figure 2B). Two-base-pair (2 bp) deletions were most commonly observed at the fourth and fifth bases from the PAM site (Figure 2C), whereas multi-base-pair deletions were noted at a variety of positions (Figure 2D). A significant observation was the frequent occurrence of one or more identical bases at the ends or edges of the multi-base deleted fragments (Figure 1C). Among the 115 cases of multi-base deletions, a substantial majority (81%) displayed identical bases at the terminus of one flanking region and at the opposite end of the deleted fragments, a phenomenon observed across all ten target sites (Figure 1D). Additionally, 7% of the deletions showed identical bases at both ends of the fragments, and 4% exhibited identical bases within their flanking regions (Figure 1C). These findings collectively suggest a unified editing mechanism for multi-base events, potentially linked to the efficacy and precision of CRISPR/Cas9-mediated genetic modification.

Amino acid sequences were inferred from the mutated nucleotide sequences, allowing for the classification of mutations into in-frame deletions (cases 1–9 in Figure 3) and frameshift mutations (cases 10–15 in Figure 3). In-frame deletions resulted in the loss of one or more amino acids, with the exception of *OsTZF5*, which showed minor in-frame deletions preceding the RR-TZF motif. These deletions did not disrupt the structural integrity of the RR-TZF motif but may be considered weak mutations. In the cases of *OsTZF8* and *OsTZF9*, in-frame deletions predominantly occurred within the spacer sequences between the two CCCH domains and also within the CCCH domain itself, where a variable number of conserved cysteine residues (C) were lost. The impact of these in-frame deletions on the functionality of the TZF motif is diverse, potentially ranging from complete disruption to mild impairment. Frameshift mutations, present in all edited genes, had the potential to result in the complete loss of the RR-TZF motif. Specifically, frameshift mutations in *OsTZF8* and *OsTZF9* led to a spectrum of truncated proteins with disrupted TZF motifs and CCCH structures.

In summary, the CRISPR/Cas9-mediated mutagenesis has successfully produced a diverse array of mutations within the rice RR-TZF genes. This valuable genetic resource lays the groundwork for functional exploration and optimization of CRISPR/Cas9 applications in rice improvement, enhancing our understanding of its mutagenic capabilities and potential for targeted genome modification.

### 2.3. Off-Target Findings in CRISPR/Cas9-Mediated Editing of Rice RR-TZF Genes

Previous studies have established that CRISPR/Cas9 possesses the potential to generate off-target mutations across diverse species [30,31,32]. Given the potential for off-target effects, we conducted a thorough analysis of T_0_ mutant plants to assess the specificity of CRISPR/Cas9 in targeting the rice *RR-TZF* genes. Potential off-target sites within the rice genome, exhibiting significant sequence homology to the sgRNAs, were identified and the occurrence of off-target events was scrutinized across one to five sites per sgRNA (Table 3).

The analysis disclosed that 5 out of 31 potential off-target sites triggered off-target mutations. Specifically, all three off-target sites for KTZF 8b, with only two base mismatches with the sgRNA, manifested mutations at rates of 76%, 8%, and 17.6%, respectively. In contrast, the presence of three base mismatches significantly reduced the likelihood of mutations, with only two out of twelve sites showing mutations at a much lower frequency, with rates of 3.6% for KTZF5 and 25% for KTZF9b. Notably, no off-target mutations were detected in the sixteen sites with four base mismatches.

These above findings highlight the substantial influence of sequence homology on off-target activity and demonstrate that the likelihood of mutation can be significantly reduced by ensuring a minimum of three base mismatches between an sgRNA and potential off-target sites.

### 2.4. Inheritance and Novel Mutations from CRISPR/Cas9 Editing

A critical aspect of functional genomics and genetic modification in rice is the heritability of induced mutations, particularly when these mutations are intended to enhance agronomic traits underpinning food security. Through self-pollination of T_0_ mutant plants, we genotyped a broad range of progeny to delineate the transmission of mutations. We self-pollinated 13 homozygous, 12 bi-allelic, and 5 heterozygous T_0_ mutant plants and performed genotyping on their progeny at the targeted sites through sequencing. This analysis encompassed 3 to 36 T_1_ plants descended from each T_0_ plant (Table 4). Our findings indicated that a complex mutation at the KTZF1 target site in the bi-allelic T_0_ mutant KT1-1 was not inherited, suggesting that it might have been chimeric. In contrast, other mutations in T_0_ plants were stably transmitted to the next generation, with new edited mutations appearing in four target sites across eight T_1_ lines.

Five homozygous double-mutant T_2_-generation lines were selected for further analysis. As detailed in Table 5, four of these lines maintained consistent genotypes with the T_1_ generation. For the KT5/6-2 line, although it did not show new edited mutations for the KTZF5 target site, it presented several new mutations for the KTZF6 target site in the T_2_ generation, underscoring the potency of CRISPR/Cas9 in targeted mutagenesis.

Additionally, we conducted genetic analysis on thirty-two progeny from the crossbreeding of double-gene-edited mutants KT5/6-2 and KT1/2-2, and the results demonstrated both stable and highly variable genotypes at the KTZF target sites (Figure 4A). KTZF1 conformed to Mendelian inheritance, whereas KTZF5 deviated, with 72% of progeny displaying T base insertions upstream of the PAM sequence, highlighting high editing activity. KTZF2 and KTZF6 also showed high editing efficiency with diverse mutation types, indicating their potential for targeted gene modification.

Given that the Cas9 protein may remain active in plant offspring, we generated sixty-eight homozygous T_2_-generation mutant lines devoid of transgenic elements, employing the selectable marker HPT (Table 4). Nonetheless, unintended mutations were still detected in the T_3_ lines (Figure 4B). These observations highlight the complexities associated with heritability and the emergence of new mutations following the application of CRISPR/Cas9 gene editing in rice, emphasizing the necessity to understand the editing characteristics and genetic stability of targeted sites.

### 2.5. Phenotypic Diversity of RR-TZF Gene Mutants in Agronomic Traits

The phenotypic consequences of CRISPR/Cas9-induced mutations in the *RR-TZF* gene subfamily were evaluated in twenty-three T_2_ homozygous mutant lines, each bearing frameshift mutations. This assessment aimed to explore the impact of these mutations on key agronomic traits, including plant height, tiller number, effective panicle number, panicle length, and seed setting rate, in comparison to the wild-type Nipponbare (WT) control. The results of this comparative analysis are presented in Table 6.

Seed setting rates in the mutants ranged from 22.6 ± 11.6% to 56.1 ± 9.7%, all of which were lower than the WT’s rate, 68.0 ± 10.3%. A significant reduction in panicle length was observed in the majority of mutants, except for lines KT6-3, KT7-1, KT7-2, and KT9-2. Tiller number increased in all mutant lines except KT4-1 and KT5-2, while the number of effective panicles showed a variable response, with reductions in most lines for *Ostzf* 1, 2, 4, 5, and 8, and increases in *Ostzf* 6, 7, and 9. Plant height, another critical agronomic trait, was generally decreased in mutants for *Ostzf* 1, 2, 4, 5, and 6, with an opposite trend in most lines for *Ostzf* 7. It is noteworthy that mutants for *Ostzf* 8 and *Ostzf* 9 did not differ significantly from the WT.

Most strikingly, phenotypic inconsistencies were observed among mutants of the same gene. In comparison to the WT, some mutants showed either reductions or enhancements in specific traits, while others were phenotypically indistinguishable from the WT. For instance, mutants of *Ostzf* 1, 4, 5, and 6 varied in plant height, while those of *Ostzf* 4 and 5 showed variable tiller numbers. The number of effective panicles also varied among *Ostzf* 2 and 8 mutants. Notably, mutants within the *Ostzf* 6 and *Ostzf* 7 lines displayed contrasting phenotypes for panicle length.

Collectively, our results indicate that *RR-TZF* gene mutants exhibit diverse phenotypes in agronomic traits under normal field conditions, with variability observed even among mutants of the same gene. This variability underscores the intricate nature of gene-editing outcomes and the influence of possible random mutations in the genetic background during tissue culture on phenotypic expression.

## 3. Discussion

The CRISPR/Cas9 system offers a promising method for creating precise and rapid modifications in plant genomes. Advances in CRISPR/Cas have further empowered researchers to alter a greater number of genes with improved efficiency [33,34,35]. In this study, we successfully induced targeted mutations at ten out of eleven sites within the rice *RR-TZF* gene family using *Agrobacterium*-mediated CRISPR/Cas9, achieving an average mutation rate of 73.8%. Notably, we achieved a high frequency of double mutagenesis (56.3%) despite the co-transformation rate being only 3.9% to 13.8%, which is comparable to previous studies [36]. This provides an alternative method to produce single and double mutants in one transformation, cutting down on the number of experiments needed and boosting efficiency. Additionally, new site-specific mutations were detected in selfed or crossed progeny, suggesting that the mutagenicity is inheritable. Thus, we have successfully generated various types of *RR-TZF* mutants including single, double, and multiplex mutants.

Previous research indicated that the CRISPR/Cas9 system typically generates double-strand breaks (DSBs) at positions three base pairs upstream of the PAM, leading to blunt ends that are prone to single-base insertions or deletions, particularly at the fourth base pair relative to the PAM sequence [37]. In this study, it was also shown that the majority of mutations occurred precisely at the fourth base pair upstream of the PAM sequence, with a predominance of single-base insertions or deletions, as depicted in Figure 2A–C. Additionally, we also identified a high proportion of multi-base deletions deviating from the anticipated DSB location and exhibiting flanking sequences with one to multiple identical bases on both sides, as illustrated in Figure 2D. Earlier findings have highlighted that a small subset of mutations occur at sites deviating from the predicted DSB location, possibly due to the accurate repair of the base pair immediately upstream of the PAM sequence during the DSB repair process [38]. However, given that a significant portion of multi-base-deletion sequences exhibited identical sequences flanking the mutation site in this study, it is possible that the mutation sites are not only associated with the accurate repair of the fourth base but also with the recognition of its identical sequences on either side of the deletion site. Further analysis is needed to elucidate this specific recognition mechanism.

In our current research, we scrutinized 31 putative off-target sites for potential editing events triggered by 10 sgRNAs. Mutations were identified in 5 of these 31 sites (Table 3). A detailed analysis correlating the number of mismatches in target-like sequences with off-target activity showed that all sequences with two mismatches and 16.7% of those with three mismatches were subject to cleavage. These findings suggest that a minimum of three mismatches between the sgRNA and potential off-target sequences is necessary to significantly reduce the likelihood of off-target effects, consistent with a previous report [39], which is of great significance for optimizing the application of the CRISPR/Cas9 system in plants. However, further research is needed to broaden the scope of detection in order to comprehensively assess the off-target risks associated with this system in plants. Fortunately, various approaches, such as enhanced CRISPR/Cas systems and novel methods, have been developed to mitigate the potential for off-target editing [40,41,42]. Recent studies have demonstrated that a PAM-less/free high-efficiency adenine base editor toolbox (PhieABE toolbox) can effectively prevent off-target editing in rice, thus offering significant potential for plant functional genomics and crop improvement [43].

Despite the promising results from CRISPR-Cas gene manipulation [44,45], it is crucial to acknowledge that while numerous CRISPR-Cas-modified genes have demonstrated improvements in rice traits, only a small number of lines have been validated through field trials. Among these, editing of *OsKRN2* stands out as a significant contributor to increased rice grain yield without observable negative effects on other agronomic traits in field trials [46]. Our study, as depicted in Table 6, highlights the potential variability in phenotypic expression among genetically modified plants under normal field conditions. The discrepancies observed in agronomic traits, particularly among mutants of the same gene, suggested that the diversity of targeted mutations, inheritance patterns, and the occurrence of novel mutations or off-target effects in CRISPR/Cas9-mediated editing as well as multiple unknown random mutations in the genome occurring during the process of tissue culture may influence phenotypic outcomes. To address these challenges and ensure a more accurate evaluation of the impact of targeted gene modifications on agronomic characteristics, multiple generations of backcrossing are essential. This process is critical not only to homogenize the genetic background but also to eliminate potentially negative effects stemming from off-target mutations, unintended novel mutations, and other tissue culture-induced genomic variations. Future research should prioritize integrating rigorous backcrossing protocols into breeding programs to maximize the reliability and agronomic utility of CRISPR-Cas9-mediated modifications.

## 4. Materials and Methods

### 4.1. Plant Materials and Growth Conditions

The *japonica* rice variety *Oryza sativa* cv. Nipponbare was selected as the recipient for transgenic manipulation in this study. Transgenic T_0_ plants were cultivated in a greenhouse environment at Fuzhou experimental station (26.08° N, 119.28° E) in Fujian Province, China. The mature seeds harvested from these T_0_ plants were germinated in darkness and subsequently transplanted into soil. The resulting seedlings were grown to maturity under standard field conditions. The phenotypic analysis of homozygous T_2_ mutant lines was conducted to assess the impact of the genetic modifications.

### 4.2. Design of sgRNAs and Vector Construction

Utilizing the online CRISPR/Cas9 target prediction tool [47], we designed single-guide RNAs (sgRNAs) to target each of the nine rice *RR-TZF* genes (*OsTZF1* to *OsTZF9*). The sgRNAs were carefully selected to minimize off-target effects, adhering to the criteria of low homology with other genomic sequences and avoiding homology within the 8- to 12-nucleotide seed sequence proximal to the PAM site. The 5′ ends of the sense and antisense strands of these sgRNAs were modified by adding the adapters CAG and AAC, respectively. After synthesis by the company and subsequent annealing, they were integrated into the plant CRISPR/Cas9 expression vector VK005-01 according to the protocol provided by VIEWSOLID BIOTECH. This vector features a codon-optimized mpCas9 gene under the control of a maize ubiquitin promoter and an sgRNA scaffold driven by the rice U6 promoter.

### 4.3. Rice Transformation and Generation of Transgenic Plants

The recombinant vectors were introduced into *Agrobacterium tumefaciens* strain LBA4404 by electroporation. The transformation of *Oryza sativa* cv. Nipponbare was performed using the *Agrobacterium*-mediated method, as described previously [48].

Transgenic T_0_ plants were identified by PCR, with genomic DNA extracted from rice leaves using the CTAB method. The PCR primers CZF (5′-GGGAGATCCAGCTAGAGGTC-3′) and CZR (5′-GGAAGGAGGAAGACAAGG-3′) were used to amplify a 536 bp fragment, with the following PCR program: 94 °C for 5 min, {94 °C for 45 s, 56 °C for 45 s, 72 °C for 45 s} for 35 cycles, and 72 °C for 8 min. The PCR products were electrophoresed on agarose gels to confirm the presence of the transgene.

### 4.4. Mutation and Off-Target Detection

Genomic DNA from T_0_ transgenic plants was subjected to PCR amplification using primer pairs that flank the target sites (Appendix A) and potential off-target sites (Appendix A). The amplified products were directly sequenced using the Sanger method. Mutations were identified by comparing the sequences of the transgenic plants with those of the wild-type (WT) plants. Homozygous mutations were determined by the presence of a single clear peak in the sequencing chromatogram, whereas heterozygous or bi-allelic mutations were indicated by overlapping peaks and were resolved by manual decoding.

### 4.5. Phenotype Analysis

The phenotypic assessment of mutants included measurements of plant height, tiller number, effective panicle number, panicle length, and seed setting rate at the maturity stage. For each plot, five to ten individual plants were randomly selected for analysis, excluding those at the periphery. The average values obtained were considered as the performance metrics for each genotype. Effective panicle number was determined by counting all rice panicles possessing at least five grains, and the average length of these panicles was recorded as the panicle length for each plant. The seed setting rate was calculated based on the ratio of filled grains to the total grain count in all effective panicles.

### 4.6. Statistical Analysis

Data collected were statistically evaluated using one-way ANOVA and Duncan’s multiple range tests, as implemented in the SPSS software (IBM SPSS Statistics, version 22, Cary, NC, USA). A *p*-value of less than 0.05 was considered to indicate statistical significance. The seed setting rate data were initially transformed using an inverse sine transformation to normalize the distribution before further analysis.

## 5. Conclusions

Our research provides a profound insight into the application of the CRISPR/Cas9 system for editing the *RR-TZF* gene subfamily in rice. The high efficiency of targeted mutagenesis, coupled with the rich diversity of induced mutations, underscores the potential of CRISPR/Cas9 as a tool for rice improvement and functional genomics. However, the presence of off-target effects and the inheritance of induced mutations emphasize the need for meticulous sgRNA design and a thorough understanding of the molecular consequences of gene editing. The phenotypic variability among mutants, particularly within the same gene, further highlights the intricate interplay between genetic modifications and phenotypic expression, advocating for multi-generation backcrossing to achieve a stable genetic platform, especially for quantitative trait assessment.

## Figures and Tables

**Figure 1 ijms-26-01354-f001:**
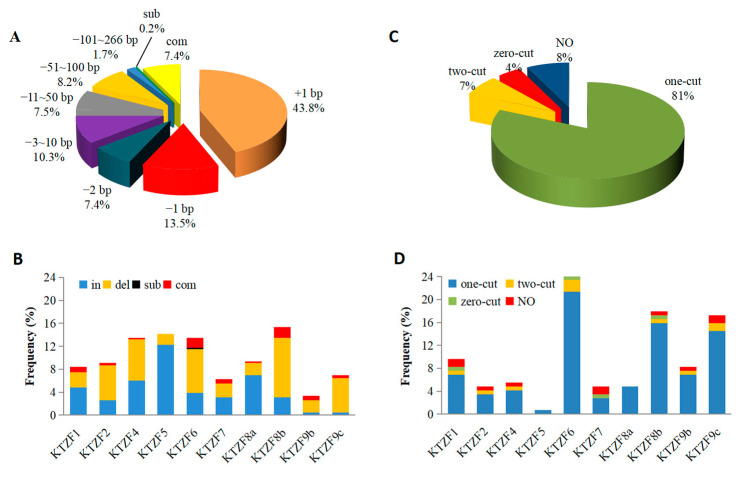
Statistics of different mutation types in target genes. (**A**) Overall frequencies of different mutation types. +/−, base insertion/deletion; sub, substitution of bases; com, complex mutation containing two or three types of base insertion, deletion, or substitution. (**B**) Frequencies of different mutation types at each target site. In, insertion of bases; del, deletion of bases. (**C**) Overall frequencies of different multi-base deletion types. one-cut, only one of the two identical sequences (located at both ends or flanking regions of the deleted fragments) has been cut; two-cut, both identical sequences have been cut; zero-cut, neither of the two identical sequences has been cut; NO, no pairs of identical sequences present at both ends or flanking regions. (**D**) Frequencies of different multi-base deletion types at each target site.

**Figure 2 ijms-26-01354-f002:**
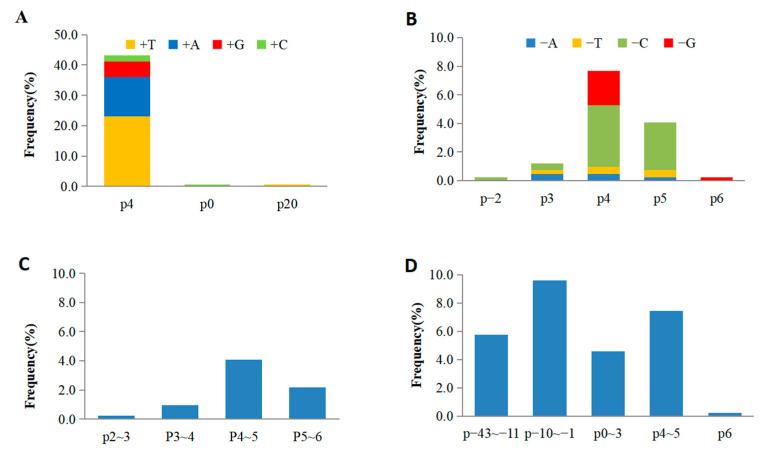
Locations and frequencies of different mutation types. (**A**) Single-base insertion. (**B**) Single-base deletion. (**C**) Two-base deletion. (**D**) Multi-base deletion. In the abscissa axis, the numbers following the letter p indicate the positions relative to the base N of the NGG site in the PAM sequence. The positive and negative numbers indicate that the positions are upstream and downstream of the base N, respectively.

**Figure 3 ijms-26-01354-f003:**
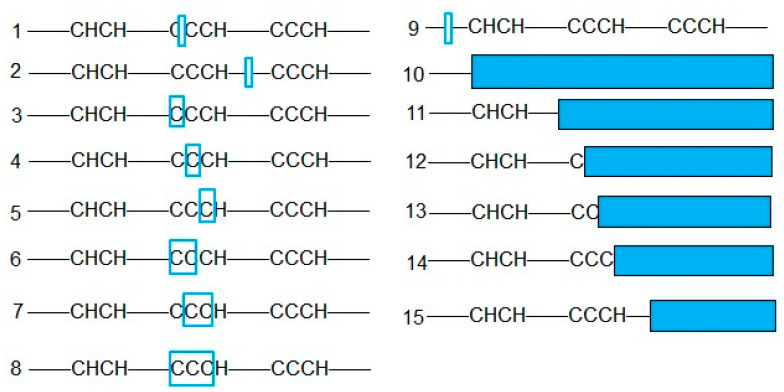
Amino acid sequence analysis of target mutant genes. Hollow boxes indicate the amino acids missed due to in-frame deletions. Solid boxes indicate the amino acids missed due to premature translation termination resulting from frameshift mutations. Cases 1–9 indicated in-frame deletions; cases 10–15 indicated frameshift mutations.

**Figure 4 ijms-26-01354-f004:**
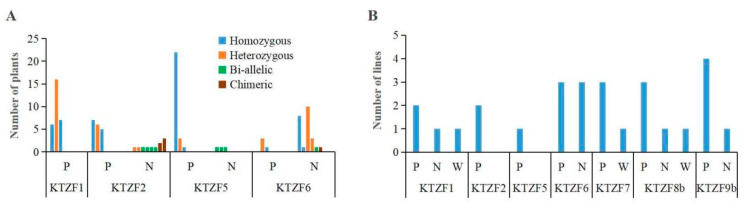
Genetic analysis of mutant progeny. (**A**) Genetic analysis of F_2_ plants from the cross between double-mutant lines KT1/2-1 and KT5/6-2. (**B**) Genetic analysis of Cas9/gRNA-negative homozygous mutant progeny. P, parental genotype; N, new mutation; W, wild-type genotype.

**Table 1 ijms-26-01354-t001:** Identification of double mutations in T_0_ transgenic plants.

Target1	Target2	No. of Transgenic Plants	No. of Co- Transformed Plants	No. of Plants with Double Mutations	Rate of Co-Transformation (%)	Rate of Double Mutation (%)
KTZF1	KTZF2	43	3	3	7.0	100.0
KTZF3	KTZF4	23	1	0	4.3	-
KTZF5	KTZF6	51	2	2	3.9	100.0
KTZF7	KTZF8a	34	4	1	11.8	25.0
KTZF8a	KTZF8b	31	2	1	6.5	50.0
KTZF9b	KTZF9c	29	4	2	13.8	50.0
Total		211	16	9	7.6	56.3

**Table 2 ijms-26-01354-t002:** Identification of targeted mutations in T_0_ transgenic plants.

Target	No. of Transgenic Plants	No. (Percentage) of Plants with Mutations	No. (Percentage) of Different Types			
			Heterozygous	Bi-Allelic	Homozygous	Chimeric
KTZF1	24	17 (70.8%)	0	10 (41.7%)	6 (25.0%)	1 (4.2%)
KTZF2	22	20 (90.9%)	0	15 (68.2%)	4 (18.2%)	1 (4.5%)
KTZF4	42	30 (71.4%)	4 (9.5%)	19 (45.2%)	7 (16.7%)	0
KTZF5	33	30 (90.9%)	1 (3.0%)	9 (27.3%)	20 (60.6%)	0
KTZF6	34	29 (85.3%)	1 (2.9%)	22 (64.7%)	3 (8.8%)	3 (8.8%)
KTZF7	24	19 (79.2%)	8 (33.3%)	5 (20.8%)	4 (16.7%)	2 (8.3%)
KTZF8a	58	27 (46.6%)	17 (29.3%)	1 (1.7%)	5 (8.6%)	4 (6.9%)
KTZF8b	39	34 (87.2%)	1 (2.6%)	21 (53.8%)	9 (23.1%)	3 (7.7%)
KTZF9b	16	7 (43.8%)	0	1 (6.3%)	6 (37.5%)	0
KTZF9c	17	15 (88.2%)	1 (5.9%)	0	13 (76.5%)	1 (5.9%)
Total	309	228 (73.8%)	33 (10.7%)	103 (33.3%)	77 (24.9%)	15 (4.9%)

Note: Heterozygous indicates that one allele is mutated and the other remains wild-type; bi-allelic and homozygous indicate both alleles are mutated with different and identical mutation, respectively; and chimeric indicates different cells or tissues in the same organism have varying mutations at the target site.

**Table 3 ijms-26-01354-t003:** Mutations detected in putative CRISPR/Cas9 off-target sites.

Target	Putative Off-Target	Sequence of the Putative Off-Target Site (5′−3′)	Putative Off-Target Locus in Chromosome	No. of Mismatching Bases	No. of Plants Sequenced	No. of Plants with Mutations	Mutation Rate (%)
KTZF1	1OTF1	GAaGCCtCCTCCGCTcgCCGTGG	chr2: 27524662 (+)	4	16	0	0
	1OTF3	GAGGaCCCCgCCGCcAgCCGCGG	chr7: 20447429 (+)	4	16	0	0
	1OTF4	GAGGCCCCCTCCGCTtcgCcCGG	chr1: 38434775 (−)	4	16	0	0
KTZF2	2OTF1	GTcCcTCGACatCGCTCACGCGG	chr1: 803496 (+)	4	17	0	0
	2OTF2	cTgCATCGAgTCaGCTCACGTGG	chr3: 6321092 (+)	4	16	0	0
	2OTF3	GTAaATCGtCTtCGaTCACGTGG	chr2: 18154523 (+)	4	16	0	0
KTZF4	4OTF1	GCgGCtCACCGAGtCGTACAAGG	chr2: 33564643 (+)	3	29	0	0
	4OTF2	GaAGtCgACCGAGCCGTACATGG	chr2: 1602272 (−)	3	26	0	0
	4OTF3	GgAGgCCACCGAGgCGTACATGG	chr3: 33963265 (−)	3	28	0	0
	4OTF4	GCAGgCCgCCGcGCCGcACACGG	chr12: 7241705 (+)	4	29	0	0
KTZF5	5OTF2	GAGGAGGAGgAGGcGTtTCTTGG	chr7: 14578311 (−)	3	27	0	0
	5OTF3	GAGGAaaAGAAGGcGTCTCTTGG	chr9: 22723220 (−)	3	28	1	3.6
KTZF6	6OTF1	TCcAGATCaCCAgGGGCATCAGG	chr1: 5216189 (+)	3	25	0	0
	6OTF2	TCGAGATCGCCAgGatCATgAGG	chr4: 29961659 (+)	4	25	0	0
	6OTF3	agGAGATCaCCAAGGGCAcCCGG	chr7: 28129157 (−)	4	25	0	0
KTZF7	7OTF1	CTgTTcGGCTGcCCgCGCTGCGG	chr1: 22815910 (−)	4	20	0	0
KTZF8a	8aOTF1	GGAGcTTCTtGCCGaGCTcCTGG	chr3: 27771740 (+)	4	23	0	0
	8aOTF2	GGAGTaaCTAGCCctGCTGCAGG	chr5: 5925375 (+)	4	23	0	0
	8aOTF3	GGAGgTTCTtGaCGCGCcGCCGG	chr6: 20946610 (−)	4	27	0	0
KTZF8b	8bOTF1	GGtCGGCGACGCaGAAGTCCTGG	chr3: 27733630 (+)	2	25	19	76.0
	8bOTF3	GGtCGGCGACGCaGAAGTCCTGG	chr5: 5965497 (−)	2	25	2	8.0
	8bOTF5	GGtCGGCGACGCaGAAGTCCTGG	chr3: 27740950 (+)	2	34	6	17.6
	8bOTF2	cGGCGGCGgCGCGGAAGgCCCGG	chr2: 34812011 (−)	3	33	0	0
	8bOTF4	GGGCGtCGAtGCGGAAcTCCGGG	chr4: 32388361 (+)	3	32	0	0
KTZF9b	9bOTF1	GaAGtGGCAACGGtGGCCGTAGG	chr1: 26286995 (+)	3	6	0	0
	9bOTF2	GcgGCGGCgACGGAGGCCGTCGG	chr4: 24069777 (−)	3	4	1	25.0
	9bOTF4	GTgGCaGCAACGGtGGCCGTTGG	chr10: 901115 (−)	3	7	0	0
	9bOTF5	GTAGCGGCAACGacGGCgGTGGG	chr11: 1945310 (−)	3	4	0	0
	9cOTF1	CCaAccaTGTCTTCGAGCTAAGG	chr4: 24207597 (−)	4	16	0	0
KTZF9c	9cOTF2	CCggTGGgGTCTTCGAGCTcGGG	chr7: 21849243 (−)	4	16	0	0
	9cOTF3	CatATGGTGTCcgCGAGCTAGGG	chr9: 17254606 (+)	4	15	0	0

Note: The PAM sequences are shown with underlines and the mismatched bases in lowercase letters.

**Table 4 ijms-26-01354-t004:** Transmission of mutations in rice *RR-TZF* genes to the T_1_ generation.

T_0_ Plant	T_0_ Genotype	T_0_ Mutation Type	No. of T_1_ Plants Tested	T_1_ Mutation Type		T-DNA-Free Homozygous Mutants from T_1_ Plants
				Identical to T_0_ Mutations	New Mutations	
KT2-2	Homozygote	−G	11	11 (−G)		7 −G)
KT5-2	Homozygote	−2	4	4 (−2)		2 (−2)
KT6-2	Homozygote	−35	5	5 (−35)		
KT6-3	Homozygote	−19	7	7 (−19)		4 (−19)
KT7-170	Homozygote	+T	4	4 (+T)		1 (+T)
KT9c-3	Homozygote	−67	6	6 (−67)		2 (−67)
KT9c-4	Homozygote	−16	3	3 (−16)		1 (−16)
KT9c-5	Homozygote	−25+30	12	12 (−25+30)		5 (−25+30)
KT1-2	Homozygote	−10	7	6 (−10)	1 (+A)	2 (−10)
KT8b-23	Homozygote	−A	7	6 (−A)	1 (+210)	1 (−A)
KT8b-76	Homozygote	−d74	36	27 (−74)	8 (+678), 1 (−74/+678)	8 (−74), 2 (+678)
KT8b-38	Homozygote	−266	18	16 (−266)	2 (−C)	6 (−266)
KT9b-2	Homozygote	−51	8	7 (−51)	1 (+A)	3 (−51), 1 (+A)
KT1-1	Bi-allelic	+T/−31+28	8	6 (+T)	2 (−56+9)	5 (+T), 2 (−56+9)
KT2-1	Bi-allelic	−2/−16	6	4 (−2), 2 (−2/−16)		4 (−2)
KT4-1	Bi-allelic	+A/−30	13	5 (+A), 1 (−30), 7 (+A/−30)		2 (+A), 1 (−30)
KT4-2	Bi-allelic	−C/−24	11	6 (−C), 3 (−24), 2 (−C/−24)		2 (−C)
KT4-3	Bi-allelic	+T/−13	11	2 (+T), 1 (−13), 8 (+T/−13)		1 (+T)
KT5-1	Bi-allelic	+T/−T	12	1 (+T), 5 (−T), 6 (+T/−T)		1 (+T), 5 (−T)
KT5-3	Bi-allelic	+T/−10	10	4 (+T), 3 (−10), 3 (+T/−10)		
KT6-1	Bi-allelic	−105/−7+14	10	6 (−105), 2 (−7+14), 2 (−105/−7+14)		6 (−105), 2 (−7+14)
KT7-1	Bi-allelic	+A/−19+T	6	1 (+A), 1 (−19+T), 4 (+A/−19+T)		
KT7-2	Bi-allelic	+C/−2	5	2 (+C), 3 (+C/−2)		
KT7-3	Bi-allelic	−G/−3	9	5 (−G), 1 (−3), 3 (−G/−3)		
KT9b-1	Bi-allelic	−21/−24	15	6 (−21), 4 (−24), 5 (−21/−24)		1 (−21), 2 (−24)
KT7-176	Heterozygote	+A	6	2 (+A)	4 (+A/+C)	
KT7-171	Heterozygote	−20	8	3 (−20), 3 (−20/WT), 1 (WT)	1 (−20/+C)	
KT8a-1	Heterozygote	−9	9	1 (−9), 4 (−9/WT), 4 (WT)		1 (−9)
KT8a-3	Heterozygote	−35	14	1 (−35), 13 (WT)		
KT8a-4	Heterozygote	−17	14	3 (−17), 3 (−17/WT), 8 (WT)		3 (−17)

Note: A “+” before a number or base indicates insertion, while a “−” indicates deletion. The number preceding the brackets shows the number of mutant plants. “WT” stands for “Wild-type”.

**Table 5 ijms-26-01354-t005:** Transmission of double mutations in rice *RR-TZF* genes from T_1_ to T_2_ generations.

T_1_ Plant	T_1_ Genotype		No. of T_2_ Plants Tested	T_2_ Genotype	
	Target 1	Target 2		Target 1	Target 2
KT1/2-1	−8/−8	−4/−4	6	−8/−8	−4/−4
KT1/2-2	−8/−8	−5/−5	6	−8/−8	−5/−5
KT1/2-3	−98/−98	−5/−5	12	−98/−98	−5/−5
KT5/6-1	+T/+T	−12/−12	7	+T/+T	−12/−12
KT5/6-2	+T/+T	+G/+G	10	+T/+T	+G/+G (3), +G/+C (1), +G/+T (1), +G/−20 (1), +G/−41 (1), +T/−20 (1), −20/−20 (1), +T/WT (1)

Note: The annotations for “+”, “−”, and “WT” are the same as in Table 4. The numbers in the brackets indicate the number of mutants with different genotypes.

**Table 6 ijms-26-01354-t006:** Agronomic traits of homozygous T_2_ mutant lines.

Target Gene	Lines	Mutation Type	Plant Height (cm)	No. of Tillers	No. of Effective Panicles	Length of Panicles (cm)	Seed Setting Rate (%)
	Nipponbare	WT	57.9 ± 2.9 c	11.8 ± 2.2 e	10.3 ± 1.4 c	14.3 ± 0.6 c	68.0 ± 10.3 a
*OsTZF1*	KT1-1	−56+9	56.5 ± 2.8 c	26.4 ± 5.3 a	9.1 ± 2.0 d	13.4 ± 0.2 e	51.6 ± 3.6 b
	KT1-2	+T	57.1 ± 4.4 c	13.8 ± 5.4 c	8.9 ± 2.0 d	13.5 ± 0.3 e	49.8 ± 7.0 b
	KT1-3	−10	50.6 ± 1.9 h	19.2 ± 6.5 b	9.4 ± 2.4 d	13.4 ± 0.3 e	55.7 ± 10.8 b
*OsTZF2*	KT2-1	−GA	54.5 ± 2.4 d	14.0 ± 2.9 c	9.8 ± 1.8 c	13.2 ± 0.5 e	52.1 ± 11.8 b
	KT2-2	−G	55.0 ± 0.6 d	13.0 ± 4.5 d	9.0 ± 1.8 d	13.8 ± 1.0 d	29.3 ± 8.2 h
*OsTZF4*	KT4-1	+A	50.4 ± 1.5 h	10.4 ± 4.3 e	7.8 ± 1.3 e	12.7 ± 0.4 j	36.8 ± 6.8 e
	KT4-2	+T	56.4 ± 1.9 c	14.4 ± 6.3 b	8.4 ± 2.6 d	12.5 ± 0.4 l	27.7 ± 5.6 i
	KT4-3	−13	55.9 ± 3.0 c	13.8 ± 4.0 c	8.8 ± 1.6 d	13.0 ± 0.5 h	46.9 ± 14.2 b
*OsTZF5*	KT5-1	−TC	56.5 ± 2.3 c	12.2 ± 1.3 d	8.1 ± 1.1 d	13.1 ± 0.3 g	52.3 ± 10.3 b
	KT5-2	+T	55.4 ± 2.6 c	10.6 ± 2.2 e	7.0 ± 1.7 g	12.8 ± 0.2 i	33.4 ± 6.8 f
	KT5-3	−10	54.2 ± 1.5 e	14.2 ± 1.9 c	7.0 ± 2.1 g	12.7 ± 0.4 j	22.6 ± 11.6 j
*OsTZF6*	KT6-1	−35	54.7 ± 3.3 d	13.0 ± 3.2 d	10.6 ± 2.5 c	13.2 ± 0.5 f	48.3 ± 8.3 b
	KT6-2	−7+14	57.3 ± 2.6 c	13.0 ± 2.5 d	11.1 ± 3.5 b	13.8 ± 0.7 d	48.8 ± 7.6 b
	KT6-3	−19	57.3 ± 3.3 c	14.2 ± 2.8 c	14.9 ± 3.3 a	14.5 ± 0.5 b	40.0 ± 7.0 d
*OsTZF7*	KT7-1	−20	63.3 ± 2.2 b	17.8 ± 3.4 b	15.3 ± 3.1 a	14.9 ± 0.5 a	40.9 ± 8.2 d
	KT7-2	+C	67.1 ± 2.5 a	15.8 ± 1.9 d	13.8 ± 2.2 a	15.2 ± 0.6 a	45.5 ± 11.9 b
	KT7-3	+T	56.4 ± 2.0 c	18.0 ± 2.6 b	14.7 ± 3.8 a	13.7 ± 0.7 e	33.4 ± 3.4 f
*OsTZF8*	KT8-1	−266	58.9 ± 1.4 c	16.0 ± 3.4 b	10.4 ± 3.4 c	12.8 ± 0.4 j	32.3 ± 9.7 g
	KT8-2	−74	58.9 ± 1.4 c	15.2 ± 2.4 b	9.2 ± 3.2 d	13.2 ± 0.5 e	56.1 ± 9.7 b
	KT8-3	+678	56.7 ± 1.7 c	13.5 ± 2.4 c	9.8 ± 2.3 c	13.5 ± 0.4 e	52.4 ± 9.8 b
*OsTZF9*	KT9-1	−67	56.8 ± 1.6 c	15.4 ± 1.8 b	10.1 ± 1.6 c	12.9 ± 0.3 i	49.7 ± 4.8 b
	KT9-2	+A	56.3 ± 1.9 c	15.0 ± 2.1 b	11.3 ± 3.7 b	14.3 ± 0.4 c	42.4 ± 9.5 c
	KT9-3	−16	56.2 ± 1.3 c	13.2 ± 2.2 d	11.2 ± 2.2 b	13.5 ± 0.6 e	49.2 ± 12.0 b

Note: The different lowercase letters indicate significant difference at *p* < 0.05.

## Data Availability

Data are contained within the article.

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
