# Peer review of "Editing the RR-TZF Gene Subfamily in Rice Uncovers Potential Risks of CRISPR/Cas9 for Targeted Genetic Modification"

_ijms, 2025, doi:10.3390/ijms26031354_

Round 1

Reviewer 1 Report

Comments and Suggestions for Authors

1, line 42-44:During cleavage site repairing, the NHEJ pathway often leads to small InDels (insertions and deletions) or substitutions, which can be utilized to elucidate gene functions and to develop novel germplasm resources [11].

Revise the logical relationship of NHEJ in elucidating gene functions.

2, Line 55-56:As plant genome sequencing projects continue to escalate, the RR-TZF gene subfamily has been systematically identified across various plant species [21].

Additionally, provide information on which species the RR-TZF gene subfamily has been identified.

3,In the first paragraph of the discussion, primarily focus on describing and analyzing your own results. Please incorporate other researchers' studies and analyze the similarities and differences compared to their findings.

4,In Line 336, please add the specific location of the greenhouse environment in the materials and methods section.

Reviewer 2 Report

Comments and Suggestions for Authors

The study reports that the CRISPR/Cas9-mediated target gene editing also leaded to a few off-target genes’ mutation by editing 8 RR-TZF genes in rice. Totally, the manuscript was well organized and easy to follow. However, although the study is of significance for researchers in this field, the manuscript is required to be improved significantly.

1.     ‘uncovers potential risks’ is not appropriate to be used in the title, because the off-targets in homologous genes have been widely reported previously. I suggest it could be changed to ‘Off-target analysis of CRISPR/Cas9 technology in editing RR-TZF gene subfamily in rice.’

2.     The abstract needs further concise and makes it more focus on the conclusions obtained in the study.

3.     P4L101, what's the difference between bi-allelic and chimeric mutations? It is best for the authors to present one example for each kind of mutation type in the table2.

4.     P10L230-236, the evidences are not enough to the conclusion in this part. Only using PCR specific to selectable marker HPT cannot confirm the transgenic elements were removed from mutant lines. Here, Southern Blotting is generally required. If you cannot make sure that the CRISPR/CAS9 was removed from the transgenic mutant lines, you cannot come to this conclusion.

5.     P11L253, this sentence is contradictory to L249-250, which stated that the reductions in tiller number in Ostzf8 but increases in Ostzf9'. Please double check it.

6.     L255-266, lots of morphological changes were observed in these genes’ knockout lines, while, what are the caused change due to the target gene mutation? Which should be presented firstly by analyzing related data; and then, move to some unanticipated changes.  

7.     In Discussion, I would like to suggest the authors put some sentences to emphasize the importance of backcrossing to eliminate possibly negative effects from CRISPR/Cas9.

Reviewer 3 Report

Comments and Suggestions for Authors

This manuscript presents a detailed investigation into the applying CRISPR/Cas9 technology for editing the RR-TZF gene subfamily in rice. The study is well-organized and offers valuable insights into various aspects of gene editing, including efficiency, mutation patterns, off-target effects, and phenotypic outcomes. Specifically, the research achieved a high editing efficiency across eight RR-TZF genes in the japonica rice variety Nipponbare. These findings provide a better understanding of the mechanisms underlying CRISPR/Cas9-mediated editing and highlight the uncertainties associated with this technology. This work serves as a valuable resource for developing future strategies to enhance cereal and rice improvement. However, this manuscript needs small improvement.

Minor Comments

  • There are a few typographical errors in the manuscript. For example, line 19, the spelling of “Additonally” must be corrected. Please check whole manuscript.

  • Add a sentence in the abstract including your result about off-target effects and phenotypic analysis.
